# The Vagus Nerve Regulates Immunometabolic Homeostasis in the Ovine Fetus near Term: The Impact on Terminal Ileum

**DOI:** 10.3390/biology13010038

**Published:** 2024-01-09

**Authors:** Mingju Cao, Shikha Kuthiala, Keven Jason Jean, Hai Lun Liu, Marc Courchesne, Karen Nygard, Patrick Burns, André Desrochers, Gilles Fecteau, Christophe Faure, Martin G. Frasch

**Affiliations:** 1Department of Obstetrics and Gynaecology and Department of Neurosciences, CHU Ste-Justine Research Centre, Université de Montréal, Montréal, QC H3T 1C5, Canada; mcly3@hotmail.com (M.C.);; 2Biotron Microscopy, Western University, London, ON N6A 3K7, Canada; 3Clinical Sciences, CHUV, Université de Montréal, St-Hyacinthe, QC J2S 2M2, Canadaandre.desrochers@umontreal.ca (A.D.);; 4Department of Pediatrics, CHU Ste-Justine Research Centre, Université de Montréal, Montréal, QC H3T 1C5, Canada; 5Centre de Recherche en Reproduction Animale, l’Université de Montréal, St-Hyacinthe, QC H3T 1J4, Canada; 6Department of Obstetrics and Gynecology and Institute on Human Development and Disability, School of Medicine, University of Washington, 1959 NE Pacific St Box 356460, Seattle, WA 98195, USA

**Keywords:** vagus nerve, immunometabolism, inflammation, fetus

## Abstract

**Simple Summary:**

The brain senses and regulates all internal bodily processes, including the levels of the sugar molecule glucose in the blood stream (systemic regulation) or in specific organs, such as the gut. The contribution of the vagus nerve to this mechanism, the longest nerve connecting the brain with the rest of the body, has not been well understood. Moreover, it has remained unknown how the activity of the vagus nerve contributes to the inflammatory response as may be seen during infections when the immune system is activated. Here, we mechanically and electrically manipulated the vagus nerve activity, decreasing or increasing it selectively, and measuring the degree of change in the glucose levels and of the inflammatory response systemically and in the gut, an important site of neonatal injury. We reported a surprising dual (hormetic) anti- or pro-inflammatory effect of reduced vagal activity combined with increased levels of glucose dependent on the levels of the infectious agent inducing the inflammatory response. We further discussed the implications for future research and clinical practice.

**Abstract:**

BACKGROUND. Glucosensing elements are widely distributed throughout the body and relay information about circulating glucose levels to the brain via the vagus nerve. However, while anatomical wiring has been established, little is known about the physiological role of the vagus nerve in glucosensing. The contribution of the vagus nerve to inflammation in the fetus is poorly understood. Increased glucose levels and inflammation act synergistically when causing organ injury, but their interplay remains incompletely understood. We hypothesized that vagotomy (Vx) will trigger a rise in systemic glucose levels and this will be enhanced during systemic and organ-specific inflammation. Efferent vagus nerve stimulation (VNS) should reverse this phenotype. METHODS. Near-term fetal sheep (n = 57) were surgically prepared using vascular catheters and ECG electrodes as the control and treatment groups (lipopolysaccharide (LPS), Vx + LPS, Vx + LPS + selective efferent VNS). The experiment was started 72 h postoperatively to allow for post-surgical recovery. Inflammation was induced with LPS bolus intravenously (LPS group, 400 ng/fetus/day for 2 days; n = 23). For the Vx + LPS group (n = 11), a bilateral cervical vagotomy was performed during surgery; of these n = 5 received double the LPS dose, LPS800. The Vx + LPS + efferent VNS group (n = 8) received cervical VNS probes bilaterally distal from Vx in eight animals. Efferent VNS was administered for 20 min on days 1 and 2 +/10 min around the LPS bolus. Fetal arterial blood samples were drawn on each postoperative day of recovery (-72 h, -48 h, and -24 h) as well as at the baseline and seven selected time points (3–54 h) to profile inflammation (ELISA IL-6, pg/mL), insulin (ELISA), blood gas, and metabolism (glucose). At 54 h post-LPS, a necropsy was performed, and the terminal ileum macrophages’ CD11c (M1 phenotype) immunofluorescence was quantified to detect inflammation. The results are reported for *p* < 0.05 and for Spearman R2 > 0.1. The results are presented as the median (IQR). RESULTS. Across the treatment groups, blood gas and cardiovascular changes indicated mild septicemia. At 3 h in the LPS group, IL-6 peaked. That peak was decreased in the Vx + LPS400 group and doubled in the Vx + LPS800 group. The efferent VNS sped up the reduction in the inflammatory response profile over 54 h. The M1 macrophage activity was increased in the LPS and Vx + LPS800 groups only. The glucose and insulin concentrations in the Vx + LPS group were, respectively, 1.3-fold (throughout the experiment) and 2.3-fold higher vs. control (at 3 h). The efferent VNS normalized the glucose concentrations. CONCLUSIONS. The complete withdrawal of vagal innervation resulted in a 72-h delayed onset of a sustained increase in glucose for at least 54 h and intermittent hyperinsulinemia. Under the conditions of moderate fetal inflammation, this was related to higher levels of gut inflammation. The efferent VNS reduced the systemic inflammatory response as well as restored both the concentrations of glucose and the degree of terminal ileum inflammation, but not the insulin concentrations. Supporting our hypothesis, these findings revealed a novel regulatory, hormetic, role of the vagus nerve in the immunometabolic response to endotoxin in near-term fetuses.

## 1. Introduction

The fetal inflammatory response is an important independent contributor to antenatal and perinatal brain injury [1,2,3]. Via the cholinergic anti-inflammatory pathway, the vagus nerve attenuates the inflammatory response to endotoxin [4]. The fetal cholinergic anti-inflammatory pathway is active in pre- and near-term ovine fetuses [5,6]. Chronically instrumented, non-anesthetized fetal sheep is an important model for human fetal development [7,8]. Using this model, we also established an approach for manipulating the fetal vagus nerve in the chronically instrumented unanesthetized fetal sheep near term [7,8].

Glucosensing elements are widely distributed throughout the body [9,10]. The carotid body, gut, pancreas, and hepatic portal/mesenteric vein have been implicated in glucosensing. From these locations, the vagus nerve carries information about circulating glucose to the brain [11]. However, while the anatomical wiring has been established, the physiological role of the vagus nerve in glucosensing, along with other key homeostatic processes, is only beginning to unravel with evidence coming from epidemiological and animal experimental studies [9,12]. 

Chorioamnionitis is an antenatal pro-inflammatory process involving the placenta and fetal membrane, which may play a contributing role in the pathogenesis of necrotizing enterocolitis of the neonate (NEC). The main manifestation of pathologic inflammation in the fetoplacental unit, chorioamnionitis, affects 20% of term pregnancies and up to 60% of preterm pregnancies [13,14]. Both symptomatic and asymptomatic chorioamnionitis are associated with an increased risk of NEC [6]. The prevention of NEC in neonates remains a major clinical challenge and a better understanding of the interplay between the synergistic contributing mechanisms of inflammation and rising glucose levels is needed [15].

The role of glycemic control in the regulation of the inflammatory response is not well understood, with evidence accruing that hyperglycemia exacerbates the inflammation induced by endotoxins [16]. Similar assumptions have been made for the etiology of NEC and other major neonatal causes of morbidity and mortality, especially in premature neonates [17,18,19,20].

Insulin treatment does not provide effective and safe glycemic control in at-risk neonates. Better therapeutic modalities for glycemic control are needed. The modulation of the vagus nerve activity may provide one such modality, but the first step must be a clearer elucidation of the role of the vagus nerve in the etiology of the fetal inflammatory response and changes in glucose homeostasis. Advancing this understanding was the goal of the present study.

In humans, Thayer and Sternberg (2006) reviewed the evidence from four studies in ~18,000 healthy subjects and subjects with diabetes type 2, in which decreased vagal function assessed via its proxy, heart rate variability (HRV) measures, was associated with increased fasting glucose and hemoglobin A1c levels. This showed overall that reduced vagal activity is a hallmark of diabetes [21,22,23,24]. Moreover, chronic cervical vagus nerve stimulation (VNS) for the control of drug-resistant epilepsy can increase fasting glucose levels, albeit still within healthy limits [25].

In animals, in contrast to some of the studies in human subjects, a recent report indicated that VNS in adult mice exposed to endotoxin reduced inflammation-induced hyperglycemia by inducing insulin [15]. The chronic hyperglycemic effects of VNS, reported in humans, were also validated in a murine study, likely due to non-selective VNS which activated the afferent fibers thought responsible for this effect, in contrast to the efferent fibers which were thought to mediate the hypoglycemic effects of VNS [26,27]. In another adult murine model study, the vagus nerve electroneurogram was shown to encode glucose levels corroborating the notion that the vagus nerve carries information about glucose homeostasis [28].

Taken together, it is evident that our knowledge is still very limited about vagal glycemic control across different species and its developmental profile. However, it is evident that the vagus nerve exerts—selectively via its efferent and afferent pathways—a powerful modulatory influence on glucose *and* inflammatory homeostatic control systems in health and disease. The evidence is also compelling that the vagus nerve’s regulatory role is present already during fetal development, a period when adverse exposures are known to have powerful long-lasting reprogramming effects on postnatal health and a predisposition to disease in later life [8,29,30].

Consequently, we hypothesized that (1) bilateral cervical vagotomy in mature, near-term, fetal sheep would cause an increase in systemic glucose levels; (2) this increase would be correlated to a higher degree of systemic inflammation and the inflammation in the terminal ileum, NEC’s locus minoris resistentiae; and (3) selective efferent VNS would reverse these patterns (glycemic control hypothesis). 

## 2. Methods

### 2.1. Ethics Approval

The animal care was performed according to the guidelines of the Canadian Council on Animal Care and the approval by the University of Montreal Council on Animal Care (protocol #10-Rech-1560).

### 2.2. Anesthesia and Surgical Procedure

The detailed approach to the chronic fetal sheep instrumentation, including the Vx and VNS procedures, was reported elsewhere [7,31]. Briefly, 57 pregnant time-dated ewes of mixed breed with singleton fetuses were instrumented at 126 days of gestation (dGA, ~0.86 gestation) with arterial, venous, and amniotic catheters; fetal chest electrocardiograms (ECG); and cervical bilateral VNS electrodes. Cervical bilateral vagotomy (Vx) during surgery was performed in 19 animals. Of these animals, eight sheep fetuses were equipped with efferent VNS electrodes and VNS treatment during the experiment in a manner reported elsewhere [7,31]. 

Both the ewe and fetus were instrumented surgically using sterile techniques under general anesthesia [7,31]. In the case of a twin pregnancy, we chose the larger fetus based on palpating and estimating the intertemporal diameter. The total duration of the surgical instrumentation was approx. 2 h. The ewe received antibiotics intravenously (trimethoprim sulfadoxine 5 mg/kg). The fetus also received antibiotics intravenously and in the amniotic cavity (ampicillin 250 mg). Warm saline was used to replace the amniotic fluid lost during surgery. The catheters and electrical wires installed in the fetus were exteriorized through the maternal flank and secured to the back of the ewe in a plastic pouch. For the duration of the experiment, the ewe was returned to the metabolic cage. In the metabolic cage, the animal could stand, lie, and eat ad libitum while we monitored the non-anesthetized fetus without sedating the mother. During postoperative recovery, antibiotic administration was continued for 3 days. Arterial blood was sampled for the evaluation of the maternal and fetal condition. The catheters were flushed with heparinized saline to maintain patency. 

### 2.3. Experimental Protocol

Postoperatively, all the animals recovered for 3 days before starting the experiments. During these 3 days, at 9.00 a.m. 3 mL of an arterial plasma sample was taken for the blood gas and cytokine analysis. Each experiment commenced at 9.00 a.m. with a 1 h baseline measurement followed by the respective intervention, as outlined below (Figure 1). The fetal heart rate (FHR) and arterial blood pressure (ABP) were monitored continuously (CED, Cambridge, UK, and NeuroLog, Digitimer, Welwyn Garden City, UK). The blood and amniotic fluid samples (3 mL) were taken for arterial blood gases, lactate, glucose and base excess (in plasma, ABL800Flex, Radiometer, Brea, CA, USA), and cytokines (in plasma and amniotic fluid) at the time points 0 (baseline), +1 (i.e., immediately after lipopolysaccharide (LPS) administration), +3, +6, +12, +24, +48, and +54 h (i.e., before sacrifice at day 3). For the cytokine analysis, the plasma was spun at 4 °C (4 min, 4000× *g* force, Eppendorf 5804R, Mississauga, ON, Canada), decanted, and stored at −80 °C for subsequent ELISAs. After the +54 h (day 3) sampling, the animals were killed as reported [7,31]. The fetal growth was assessed by the body, brain, liver, and maternal weights. 

We used the model of LPS-induced inflammation in fetal sheep. It is a well-established model of the human fetal inflammatory response to sepsis [1,2,32,33]. 

The experimental groups consisted of the three following categories.

1. Control and LPS groups: Fifteen fetuses were used as the controls receiving NaCl 0.9%. Twenty-three fetuses received LPS (100 n = 2, 200 n = 1, 400 n = 15, or 800 n = 5 as ng/fetus/day) derived from *E. coli* (Sigma L5293, from *E. coli* O111:B4, a readymade solution containing 1 mg/mL of LPS), which was administered intravenously to the fetuses on days 1 and 2 at 10.00 am to mimic high or intermittently high levels of endotoxins in fetal circulation over several days as it may occur in chorioamnionitis. As we identified that the IL-6 response did not depend on the LPS dose in the applied range [33], these animals were all considered as one LPS group for statistical comparison purposes.

2. Vx + LPS groups: Eleven animals were vagotomized (Vx) and exposed, similar to the LPS group, to LPS400 (n = 6) or LPS800 (n = 5).

3. Efferent VNS group: Eight additional Vx animals were subjected to bilateral cervical VNS applied via Digitimer NeuroLog’s NL512/NL800A using a pulse sequence pre-programmed in Spike 2 for 10 min prior to and 10 min after each injection of LPS. The VNS settings were as follows: DC rectangular 5 V, 100 uA, 2 ms, 1 Hz according to [4]. VENG was recorded at 20,000 Hz [34]. 

### 2.4. Cardiovascular Analysis

The systolic, diastolic, and mean ABP (sABP, dABP and mABP) and the mean FHR were calculated for each animal at each time point (baseline, 1 h, 3 h, 6 h, 24 h, 48 h, and 54 h) as an average of the artifact-free 30 preceding minutes (60 preceding minutes for the baseline) using Spike 2 (Version 7.13, CED, Cambridge, UK).

### 2.5. Cytokine Analyses

The cytokine concentrations (IL-6) in the plasma were determined by using an ovine-specific sandwich ELISA. Mouse anti-sheep monoclonal antibodies (capture antibody IL-6, MCA1659, Bio Rad AbD Serotec, Montreal, QC, Canada) were pre-coated at a concentration of 4 µg/mL on an ELISA plate at 4 °C overnight. After they were washed three times with a washing buffer (0.05% Tween 20 in PBS, PBST), the plates were then blocked for 1 h with a 1% BSA in PBST. Following the three washes, 50 µL of the serial diluted protein standards and samples were loaded per well and incubated for 2 h at room temperature. All the standards and samples were run in duplicate. Recombinant sheep proteins (IL-6, Protein Express Cat. no 968-305; TNF-α, Cat. no 968-105, Shelton, CT, USA) were used as the ELISA standard. The plates were then washed three times. Rabbit anti-sheep polyclonal antibodies (detection antibody IL-6, AHP424, Bio Rad AbD Serotec) at a dilution of 1:250 were applied in the wells and incubated for 30 min at room temperature. The plates were then washed with a washing buffer five times. Detection was accomplished by assessing the conjugated enzyme activity (goat anti-rabbit IgG-HRP, dilution 1:5000, Jackson ImmunoResearch, Cat. No 111-035-144, West Grove, PA, USA) via incubation with TMB substrate solution (BD OptEIA TMB substrate Reagent Set, BD Biosciences Cat. No 555214, Franklin Lakes, NJ, USA). The color development reaction was stopped with 25 µL of 2 N sulfuric acid. The plates were read on an ELISA plate reader at 450 nm, with a 570 nm wavelength correction (EnVision 2104 Multilabel Reader, Perkin Elmer, Waltham, MA, USA). The sensitivity of the IL-6 ELISA was 16 pg/mL. For all the assays, the intra-assay and inter-assay coefficients of variance were <5%.

### 2.6. Immunofluorescence Analyses of the Terminal Ileum

Briefly, we quantified the CD11c+ expression (i.e., M1 macrophages) normalized for the CD11c+ cell counts, as published [35]. 

Terminal ileum tissue samples of approximately 10 cm in length were taken from the fetus during necropsy and immediately immersed in 4% paraformaldehyde (PFA) for 48 to 72 h. The tissue samples were then washed and stored in 1× phosphate-buffered saline (PBS) and changed daily for 3 days. Finally, the samples were stored in 70% ethanol until further processing and kept at 4 °C when they were in liquid. After that, they were processed using the Leica TP 1020 Automatic Tissue Processor (Leica Instruments, Nussloch, Germany) and then embedded in paraffin at the Leica EG 1160 Paraffin Embedding Center (Leica Microsystems, Nussloch, Germany). Using the Leica RM2145 Rotary Microtome (Leica Microsystems, Nussloch, Germany), 5 μm slices were obtained from slicing the embedded tissue samples and mounted on the Fisherbrand Colorfrost Plus microscope slides (Fisher Scientific, Waltham, MA, USA).

To detect the M1 macrophages, we stained the ileum tissues with a mouse anti-sheep monoclonal antibody CD11c (1:50 dilution; RTI, LLC, Brookings, SD, USA; Cat. no CD11C 17–196). 

All the slides were stained simultaneously with the same solutions to minimize staining variations. The slides were deparaffinized with three 5-min washes in xylene, rehydrated for 2 min each through a descending ethanol series (100%, 100%, 90%, 90% and 70%), then rinsed in deionized water for 5 min. The tissue sections were then subjected to antigen retrieval in 10 mM of sodium citrate at a pH of 6.0 in a pressurized antigen retriever (Electron Microscopy Sciences, Hatfield, PA, USA) followed by three phosphate-buffered saline (PBS) rinses at 5 min each. Then, all the subsequent steps were performed in a closed humidity chamber. Non-specific protein binding was blocked using the Background Sniper protein blocker (Biocare Medical, Pacheco, CA, USA) for 10 min, then the slides were incubated overnight at 4 °C with the respective primary antibody as outlined above. Following three further PBS washes, the slides were incubated for 40 min at room temperature using Alexa 568 goat anti-mouse IgG (Thermo Fisher, Waltham, MA, USA). After another 3 × 5-min PBS rinses, a 2-min counterstain with DAPI, and a further PBS rinse, the slides were cover-slipped using Prolong Gold mounting media (Thermo Fisher/Invitrogen, Waltham, MA, USA). 

The negative controls were performed by replacing the primary antibody step with purified pre-immune mouse IgG to rule out non-specific binding. 

Imaging and Analysis: The quantification was performed on six randomly selected high-power fields (40x Oil magnification). The images were captured using a Zeiss AxioImager Z1 microscope (Carl Zeiss Canada, Toronto, ON, Canada). Identical illumination settings were used for all the images, and the analysis was performed using the Image Pro Premier 9.2 software (Media Cybernetics Inc., Rockville, MD, USA), with the analyst blinded to the animal group. 

The combined pictures were used to quantify the CD11c+ macrophages, as described below. The complete tissue was scanned, and six to seven individual pictures were cropped for quantification using Image Pro Premier. 

We evaluated the terminal ileum tissues for the presence of CD11c+ cells in two ways. First, we quantified the density of the intensity of the CD11c+ stains (area of the stains divided by the total area of the individual HPF, “range mean”) using the Image Pro Premier 9.2 software (version 9.2; Media Cybernetics, Rockville, MD, USA). 

Second, we counted the number of CD11c+ cells as follows. After preliminarily testing of a random sample of images (including screening against negative controls), the binary intensity thresholds and settings were established to select and count only the cells and cell fragments deemed to be positively stained. Semi-automated macros were then written in the analysis software to minimize bias and ensure consistency.

Since fetal gut tissue is full of erythrocytes and other autofluorescent components, a strategy to eliminate the counting of any non-specific fluorescence was developed. As autofluorescence appears in both the green and red channels, while our true CD11c signals only appeared in the Alexa 568/red channel, we used the green channel to create a binary “exclusion” mask. This was achieved as follows. Greyscale versions of the green channel were subjected to automated, binary thresholding using the minimum variance “auto bright” algorithm in Image Pro. A count was performed to create outlines of those counted spots (mostly erythrocytes). The counted spots were grown by seven pixels to encompass any convolved signal immediately surrounding them. The outlines of these counted zones were then geographically superimposed onto the corresponding multichannel image. Following this, a Boolean exclusion was imposed, so that subsequent counting of the red CD11c signal would only take place in the regions of interest outside of the autofluorescent cellular area.

After imposing this region of interest selection, the multichannel image was slightly sharpened using a high Gaussian filter to yield better morphological data. To select and measure the CD11c-positive cells, a binary threshold was created using the “smart selection” tool in Image Pro, which selectively recognized the red cells based on the color, morphology, and tissue background characteristics. CD11c-positive cellular debris seemed to be evident in several animals, so we chose to analyze the cells and debris or “spots” separately. The cells were considered to be any objects above 350 pixels in size. The data from the debris was not included in the morphological statistics. All the cell and spot measurements were normalized to the tissue area. To determine this area, a manually created binary intensity threshold was used to identify and measure the dark background surrounding tissue, and this area was subsequently subtracted from the total image area to yield a measurement of the total tissue area in the image.

Finally, for each animal, the density of intensity of the CD11c+ cells per HPF determined in the first step above was normalized for the number of cells per HPF computed in the second step by calculating a ratio of the two. *The latter ratio was used for the statistical analyses, reported for simplicity as CD11c+ cells.*

### 2.7. Insulin Analyses

The insulin concentrations in the fetal sheep plasma were determined by using an ovine-specific insulin ELISA kit (ALPCO Diagnostics, Cat. No. 80-INSOV-E01, Salem, NH, USA). Briefly, fetal sheep blood samples were collected using a heparinized syringe at baseline, 3 h, and 48 h. Plasma samples were obtained by centrifugation at 4000× *g*, 4 °C for 4 min, and stored in aliquots at −80 °C until the assay. Twenty-five (25) microliters of plasma were loaded per well without dilution in duplicates. The sensitivity of the assay was 0.14 ng/mL. The intra-assay and inter-assay coefficients of variance were <5.96% and <5.78%, respectively.

### 2.8. Statistical Analysis

General linear modeling (GLM) in Exploratory/R was used to assess the effects of the treatment (LPS, Vx, Vx + efferent VNS + LPS) while accounting for repeated measurements of the fetal blood gases, glucose, lactate, acid–base status, insulin, fetal cardiovascular responses (blood pressure and heart rate), fetal systemic inflammatory response (plasma IL-6 cytokine), and fetal terminal ileum inflammatory response (CD11c+ cells). Consequently, the experimental groups and time points served as the predictor variables, and the base levels served the control group and the baseline time point, respectively. The coefficients of the estimates were used to rank the statistically significant differences by their magnitude (i.e., treatment effect larger or smaller) and direction (i.e., increase or decrease). Not all the measurements were obtained in each animal. In such a case, the sample size was reported explicitly.

All the results were presented as the median ± IQR (interquartile range) with a significant difference assumed for *p* < 0.05. Statistically significant correlations were reported for Spearman R2 > 0.1.

## 3. Results

### 3.1. Blood Gas and Metabolites

This experimental fetal sheep cohort’s morphometric, arterial blood gases, acid–base status, cardiovascular characteristics, and cytokine responses were reported in part for the control and LPS groups as well as for the acute post-Vx effects [7,33]. Briefly, the fetal arterial blood gases, pH, base excess (BE), and lactate were within the physiological range during the baseline in both groups. 

We reported here, for the first time, the findings in the Vx and efferent VNS groups over the 54-h course of the low-dose LPS-triggered fetal inflammatory response (Table 1). The respective baseline measurements in the groups of the vagotomized animals did not differ. We reported changes in glucose in a dedicated subsection below. Table 1 shows some detectable yet physiologically not meaningful treatment groups and time effects for the pH, pO2, pCO2, O2Sat, and BE (*p* values ranging from <0.001 and 0.3; statistically significant changes are identified in Table 1). Of note, there was a ~40% elevation in lactate in all the LPS-treated groups regardless of vagus nerve manipulation at 3 and 6 h post-LPS, corresponding to the peak of the fetal inflammatory response and indicating mild metabolic acidemia.

### 3.2. Cardiovascular Responses

Overall, we observed no overt cardiovascular decompensation due to the LPS-triggered inflammatory response (Table 2). There was a consistent 6-h time effect for the FHR and ABP corresponding to the acute fetal inflammatory response after the first dose of LPS, regardless of the treatment group (LPS without or with vagus nerve manipulation).

Specifically, for the FHR, there were no significant treatment effects, but there was a mild heart rate increase without overt tachycardia observed as a time effect for 6 (*p* < 0.001) and 48 h (*p* = 0.03), which corresponded to the immediate 6-h post-first dose of LPS administration and 24 h after the second dose. The 6-h effect was twice as pronounced than the 24 h effect.

For the dBP, there was a mild decrease due to the LPS treatment alone (*p* = 0.01), but not due to vagus nerve manipulation. There was also a similar mild decrease as a time effect at 6 and 30 h (*p* = 0.003 and *p* = 0.046, respectively), i.e., 6 h post the first and second LPS doses.

For the mBP, there was only a mild decrease as a time effect at 6 h regardless of the treatment (*p* = 0.009).

For the sBP, we observed again a time effect-related mild decrease at 6 h (*p* = 0.016) and treatment-related effects with mild increases for the Vx + LPS800 (*p* = 0.02) and efferent VNS groups (*p* = 0.046).

### 3.3. Systemic Inflammatory Response: IL-6

LPS provoked a systemic inflammatory response with a time effect, i.e., peaking at 3 and 6 h (both *p* values < 0.001), as measured by the fetal arterial IL-6 concentrations over 3 days (Figure 2). Surprisingly, Vx + LPS400 restored the levels of IL-6 to the control levels, while treatment effects with elevated IL-6 were observed for the Vx + LPS800 (*p* < 0.001), LPS (*p* < 0.001), and efferent VNS (*p* = 0.003) groups in a decreasing order of magnitude. 

Of note, in the efferent VNS group, the inflammatory response abated quicker than in the other LPS-exposed groups, while in the Vx + LPS800 group, the response persisted the longest up until 54 h post-LPS, the last experimental measurement time point. 

Last, we observed a clearly diametrical temporal profile of the fetal inflammatory response to LPS in the Vx + LPS400 versus Vx + LPS800 group, suggesting a sigmoid functional relationship between the vagal cervical denervation and the LPS dose-dependent magnitude of the fetal inflammatory response. 

### 3.4. Effects of Vagus Nerve Manipulation on the Fetal Systemic Arterial Glucose and Insulin Levels

For glucose, we found a treatment effect for both Vx + LPS400 and Vx + LPS800 (Table 1). Since there was no significant difference between the Vx + LPS400 and Vx + LPS800 insulin or glucose values at the respective time points, we combined them into one Vx + LPS group for further analysis in this subsection. There was only one positive pronounced effect for the treatment in the Vx + LPS group (*p* < 0.001), corresponding to a ~30% rise in the glucose values throughout the experiment in this group (Figure 3). That is, the rise was observed even prior to the first LPS exposure at the baseline measurement.

For insulin, we found positive and similar effects for the time at 3 h (*p* < 0.001) and for treatment in the Vx + LPS group (*p* = 0.009) and the efferent VNS group (*p* = 0.015). In the Vx + LPS group, the rise was ~2.3-fold at 3 h and ~1.3-fold in the efferent VNS group.

Ctrl, Control (no LPS, no Vx, vagotomy); Vx + LPS, bilateral cervical vagotomy during surgical instrumentation. The groups Vx + LPS400 and Vx + LPS800 were combined here due to a lack of differences in glucose and insulin behaviors with either LPS dose. The LPS of 400 or 800 indicated the respective intravenous dose of LPS in ng/fetus/day given after the baseline and 24 h later. The efferent VNS group received Vx and LPS400 and the Vx + LPS400 group was followed by VNS treatment around the LPS administration at days 1 and 2. 

The above findings showed that an intact vagus nerve activity modulated glucose homeostasis under the conditions of an LPS-induced inflammatory response. To test this further and explore whether the insulin rise had direct anti-inflammatory effects, we performed group-wise correlation analyses of glucose, insulin, and IL-6 combining the 3 h and 48 h time points, i.e., the time points when these biomarkers reflected the evolving inflammatory response. 

First, glucose correlated positively to insulin in all the experimental groups except for the LPS and Vx + LPS800 groups. Interestingly, the highest values of correlation were observed in the control, followed by the efferent VNS groups (Spearman R2 = 0.47 and 0.29, respectively). The Vx + LPS400 group showed Spearman R2 = 0.22. 

Second, we observed no correlations between IL-6 and insulin in any experimental group. 

Together, the findings suggest a feed-forward relationship between insulin and glucose that is disrupted and reduced by vagal denervation but has no direct association between insulin and IL-6. Under the conditions of Vx, LPS-triggered fetal systemic inflammatory and glycemic responses act synergistically, in part, in an LPS-dose-dependent manner.

### 3.5. Effects of Fetal Vagus Nerve Manipulation on the Terminal Ileum’s Inflammatory Response to LPS

The CD11c+ cells (M1 macrophages’) were increased in the LPS group (main effect LPS, *p* = 0.003), as demonstrated in an earlier report [35], as well as in the Vx + LPS800 group (*p* = 0.001), but not in the Vx + LPS400 or efferent VNS groups (Figure 4). 

## 4. Discussion

We present the findings on the effects of a mechanical definitive abrogation of the vagal signaling, the key neural substrate of brain–body communication, including the interruption of the cholinergic anti-inflammatory pathway, via the bilateral cervical vagotomy. This disruption alters the fetal systemic and gut regional immunometabolic responses to endotoxin exposure. The findings are summarized in Table 3 to facilitate the systematic interpretation.

### 4.1. General Expected and Novel Effects of Vagus Nerve Manipulation

Despite some mild changes, our experimental cohort’s morphometric, arterial blood gases, acid–base status, and cardiovascular characteristics were within the physiological range and representative of late-gestation fetal sheep as a model for human fetal development near term [36,37]. The effect of the low LPS dose we administered on the arterial blood gases, acid–base status, and cardiovascular responses was compatible with mild septicemia (mild compensated metabolic acidemia and hypoxia) evidenced by a transient rise of IL-6 at 3 h without overt shock with cardiovascular decompensation. The organ-specific effect of this inflammatory insult was evidenced by the rise of M1-type macrophages in the terminal ileum. These systemic and organ-specific findings were in line with what was expected from the fetal inflammatory response to the endotoxin challenge and the efferent VNS diminished these effects, again, as was expected per our understanding of the function of the fetal cholinergic anti-inflammatory pathway [5,6].

In fetal sheep near-term, despite the high degree of maturity of the autonomic nervous system, Vx did not cause a compensatory rise in the activity of the sympathetic nervous system [38]. This was also evident in the present study with no rise in FHR or dABP measured. As such, we did not attribute the observed rise in glucose levels to the effects of the sympathetically mediated stress pathway.

Beyond these expected effects of vagus nerve manipulation, the present study confirmed our glycemic control hypothesis. A complete withdrawal of peripheral vagus innervation resulted in a 72 h delayed onset of a sustained increase in systemic arterial glucose levels that began *prior* to endotoxin exposure and persisted unchanged for at least 54 h. Notably, this behavior was accompanied by transient hyperinsulinemia at the peak of the systemic inflammatory response, and, unexpectedly, LPS-dose-dependent biphasic lower or higher levels of systemic and gut inflammation. An intermittent efferent bilateral cervical VNS restored these changes. 

In summary, this study revealed three novel facets of the vagal control of inflammation and metabolism in the near-term, physiologically mature, fetus: (1) the short-term immunometabolic regulatory role, (2) the hormetic behavior and, concurrently, (3) adaptive, longer-term, predictive computation manifesting as immunometabolic memory of the former exposure to endotoxin. 

In the following, we discuss these observations and propose future experiments.

### 4.2. Immunometabolic Effects of Vagus Nerve Manipulation

Vx caused a chronic rise in glucose levels, notably, starting prior to and regardless of LPS, while efferent VNS restored this state to the control levels. Meanwhile, Vx also caused transient hyperinsulinemia at the peak of the LPS-triggered IL-6 increase, but not at the baseline. Efferent VNS restored this in part. The group-wise positive glucose–insulin correlations decreased from the control to the efferent VNS and Vx + LPS400 (but not LPS or Vx + LPS800) groups. Together, these findings demonstrated that Vx disrupts glucose–insulin homeostasis. The fact that the efferent VNS restored the hyperinsulinemia only partially indicated the synergistic effect of inflammation on the disruption of glucose–insulin homeostasis when vagal control is absent or diminished. Despite these observations, we did not observe any clear correlations between the glucose levels and M1 macrophage activity in the terminal ileum or between insulin and IL-6 or insulin and the M1 macrophage activity in any experimental group. With the caveat in mind that correlation is not causation, this could indicate that the triggered rise in glucose levels was not sufficient to cause injury. Alternatively, the mechanisms of hyperglycemic injury and the anti-inflammatory effects of insulin could involve causal pathways that were not captured in the present study. 

Anorexic or food intake-increasing signals are those that predict future energy requirements [39]. The emphasis on “prediction” is of great importance as it recapitulates the notions of the predictive brain put forward by Peters et al. in other system regulatory contexts [40]. Signals that predict increased fuel demand, e.g., cold exposure, increase both hunger and energy expenditure. This line of thought warrants further development in the context of neurally integrated immunometabolic control.

In such a framework, does inflammation signal metabolic demand that is controlled under physiological conditions by insulin via the vagus nerve, obscuring the net increase in insulin secretion? Under the disrupted conditions of Vx, inflammation would trigger gluconeogenesis and increase systemic glucose availability. Due to a lack of diminished central feedback on the beta cells/liver’s insulin production via the vagus nerve, we observed hyperinsulinemia. This was supported by the literature, reporting that VNS triggers insulin production and a reduction in the glucose levels [41,42]. Afferent VNS fibers, activated by non-selective VNS, are thought to produce chronic hyperglycemic effects, while the VNS of strictly efferent fibers mediates hypoglycemic effects [26,27]. In the present study, strictly efferent VNS was performed, and the findings aligned with the literature.

Reviewing the evidence of the past ~15 years, Meyers et al. proposed most recently that the brain systems that control various peripheral metabolic systems, such as the activities of the adipose tissues and glucose control systems, should be viewed, together, as the components of a larger, highly integrated, ‘fuel homeostasis’ control system [39]. This view places the vagus nerve with its ubiquitous distributed afferent sensing and efferent fibers at the center of this fuel homeostasis control system. The authors note that the developmental aspect of the adult metabolic phenotype requires further elucidation as there is little doubt that it plays a key role in the puzzle of metabolic function in health and disease throughout the human lifespan.

There is also a recognition that immune and metabolic regulatory systems are two sides of the same coin; an integrated, adaptive, multi-scale, molecular, and systems-level network [7,43,44]. This understanding has led to characterizing peripheral macrophages and, increasingly, also the central nervous system’s glia by their immunometabolic phenotype in response to various stressors [45,46,47,48,49,50,51]. 

Specifically for this developmental stage and its animal model, we proposed a scale-invariant concept in which modeling the integrated system-level responses to the manipulation of vagus nerve signaling under the conditions of repeated endotoxin exposure was akin to modeling the behavior of glial cells in vitro in response to the manipulation of the α7 nicotinic acetylcholine receptor (α7nAChR) under the conditions of single or multi-hit LPS exposure (Figure 5) [7,43]. In such a framework, the response pattern symmetry (or invariance) across the scales of organization, from cellular to system, appeared to be evident with regard to the changes in immunometabolic behavior. From the teleological evolutionary standpoint, such a phenomenon appears to provide adaptive benefits to complex organisms at all scales of the organization by optimizing the available and predicted energy utilization and entropy production rates via immunoceptive inference, which follows the free energy principle [52,53,54].

As a methodological limitation, we point out that we measured insulin at three time points only. This limited the conclusions on the temporal dynamic relationship between insulin and other biomarkers and requires further studies with a higher temporal resolution of insulin in relation to glucose and IL-6, in particular, in response to an inflammatory stimulus such as LPS. Given the potential of insulin to reduce inflammation, a deeper understanding of the physiology and pathophysiology of this dynamic relationship is of great translational interest.

More long-term, there is a well-established need for preventive early-life interventions to stem the epidemics of obesity [55]. Our findings suggest that VNS may emerge as one such early, postnatal intervention to reset the “metabolic imprinting”. The effect of VNS was delayed at 48 (but not yet at 3) hours. That suggests that chronically there may be a resetting influence for VNS as a treatment. This needs to be explored further.

### 4.3. Hormesis

We reported an increase in the M1 macrophage activity in the terminal ileum in the LPS group [35]. Here, we demonstrated that a complete withdrawal of vagal innervation followed by LPS exposure restored the M1 activity to the control levels. Efferent VNS had a similar anti-inflammatory effect. Of note, we observed a consistent LPS-dose-dependent hormetic behavior under Vx for both the systemic inflammatory response pattern and the regional terminal ileum inflammatory response. 

The hormetic behavior is represented by a functional dependency of anti- or pro-inflammatory response to endotoxin depending on its dose. This was captured in the visual abstract of the present work. Vx followed by LPS400, but not by the LPS800 regimen, reduced the systemic and gut-specific inflammation levels. That endotoxin dose–response pattern was compatible with the hormetic behavior of complex systems and was thought to reflect an adaptive or computationally predictive response observed widely in biological systems [56]. This was the first demonstration of such behavior under conditions of an ablated cholinergic anti-inflammatory pathway.

Remarkably, the immunometabolic memory of endotoxin exposure is reflected in the hormetic behavior. The surprisingly anti-inflammatory effect of Vx + LPS400 was even more pronounced at the second endotoxin exposure (at 24 h of the experiment) and was contrasted by the absence of any habituation for the Vx + LPS800 group. This was accompanied by higher levels of regional inflammation in the terminal ileum in this group. The efferent VNS appeared to boost the memory effect for the Vx + LPS400 group. As this finding was surprising and animal numbers were already planned for, we were unable to test this by adding a Vx + LPS800 + efferent VNS treatment, as was done for the Vx + LPS400 group. As such, the following remains an important follow-up study. Does efferent VNS in Vx + LPS800 fetuses restore the memory effect boosting habituation to the second endotoxin exposure? To further explore the discovered hormesis, systematic studies are needed on the endotoxin dose–response relationship depending on the maturity and species type.

### 4.4. Implications for NEC Etiology and Avenues of Treatment 

Impaired mesenteric oxygenation due to poor perfusion or cyanosis is considered the major predisposing factor for NEC. While the etiology of NEC in term infants remains unknown, the contribution of chorioamnionitis, often subclinical, has been discussed as one of the contributing factors. A common event is the exaggerated inflammatory response to an exogenous trigger as a prequel to NEC. This leads to increased intestinal permeability and mucosal injury [57,58]. Higher levels of cytokines, such as IL-6, can correlate with the severity of NEC in preterm neonates and contribute to chemotaxis [59,60,61].

Both short-term (hours) and chronic hyperglycemia and hypoinsulinemia promote organ injury and inflammation [62]. Our findings showed that this relationship depends on intact vagus nerve signaling. The acute, transient rise of insulin in the vagotomized animals at 3 h post-LPS was not fully recovered by efferent VNS, indicating the synergistic role of inflammation and hyperglycemia in this response.

Counterintuitively, it appears plausible that lowering the cholinergic tone in conditions of neonatal hyperglycemia and/or mild, but not severe, septicemia can help prevent NEC or mitigate its deterioration. Future studies should explore the predictive potential of monitoring and manipulating the vagal activity in neonates, perhaps via heart rate variability (HRV) monitoring or via direct non-invasive vagus electroneurograms. 

## 5. Conclusions

The key translational implications of our findings for neonatal healthcare were the potential of selective efferent VNS to reduce acute systemic and regional gut inflammation as well as the metabolic effects of the efferent VNS. The latter appeared to have complex effects on several time scales. Acutely, the metabolic effects may aid in reducing inflammation, boosting the vagally mediated immunometabolic regulatory networks. Chronically, the effects may be in resetting the in-utero programming of the susceptibility to obesity. The latter aspects were not the subject of the present study but appear to be a logical speculative extension of the findings and require further dedicated experiments. 

The surprising hormetic, LPS-dose-dependent, systemic, and ileum anti-/pro-inflammatory effects of Vx warrants further studies. Whether Vx alone or its synergy with the LPS-induced inflammatory response is responsible for the observed metabolic behaviors also requires further studies. The observed rise of glucose levels already *prior* to the inflammatory response does indicate a direct glycemic control function of the vagus nerve in near-term fetal sheep. The intermittent rise of insulin concomitant with the rise of IL-6 suggested a synergistic immunometabolic pattern.

Our results are relevant for a broader understanding of immunometabolic programming in the perinatal stage of development and the possibility of using VNS to reset it REF. The implications for understanding the involvement of the vagus nerve in glucosensing and inflammation warrant further investigation in general, particularly regarding the therapeutic opportunities for NEC.

## Figures and Tables

**Figure 1 biology-13-00038-f001:**
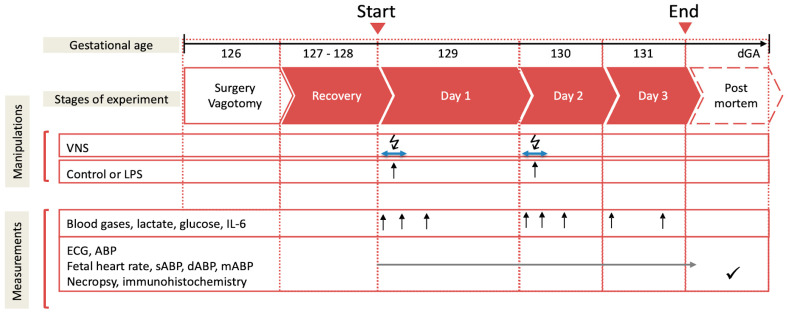
Experimental design. Bilateral cervical vagotomy (Vx) was performed during surgery in the Vx + LPS group animals. On days 1 and 2, the Vx + LPS animals received a lipopolysaccharide (LPS) dose of 400 or 800 ng/fetus/day. Some Vx + LPS animals also received the efferent intermittent (to the periphery) VNS on days 1 and 2 (efferent VNS group). Control group (n = 15), LPS group (n = 23), Vx + LPS group (n = 11), Vx + LPS + efferent VNS group (n = 8).

**Figure 2 biology-13-00038-f002:**
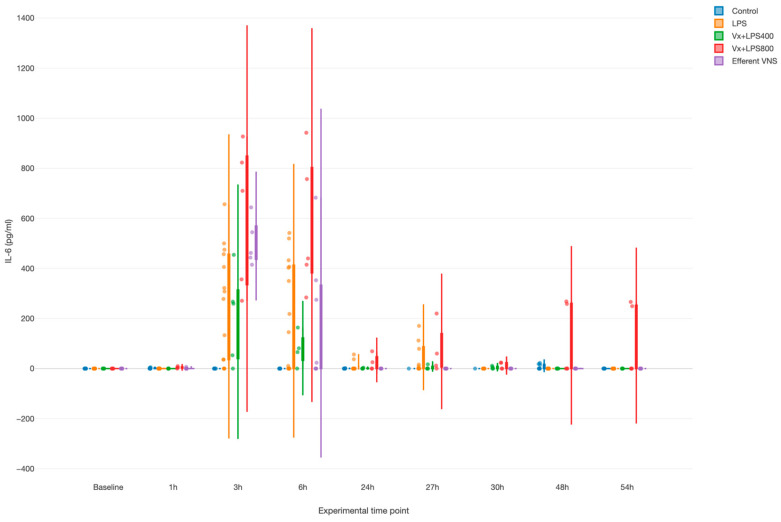
Fetal systemic inflammatory response to intravenous lipopolysaccharide (LPS) injection after the baseline and at 24 h: the impact of vagus nerve manipulation. Vx, bilateral cervical vagotomy during surgical instrumentation; LPS400 and LPS800 indicate the respective intravenous dose of LPS in ng/fetus/day given after the baseline and 24 h later; the efferent VNS group received Vx and LPS400 as the Vx + LPS400 group was followed by VNS treatment around the LPS administration at days 1 and 2.

**Figure 3 biology-13-00038-f003:**
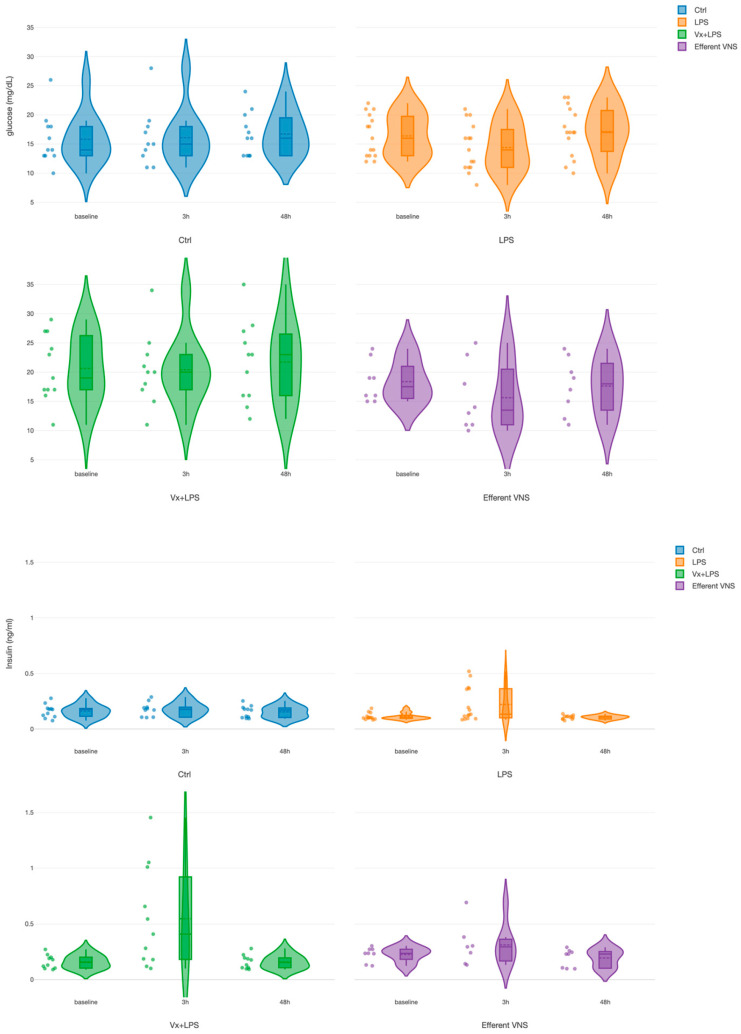
Fetal systemic arterial glucose (TOP) and insulin (BOTTOM) dynamics during the fetal systemic inflammatory response to intravenous lipopolysaccharide (LPS) injection: the impact of vagus nerve manipulation. Key time points are shown as follows: baseline (prior to LPS injection) and 3-h and 48-h post-first LPS injection.

**Figure 4 biology-13-00038-f004:**
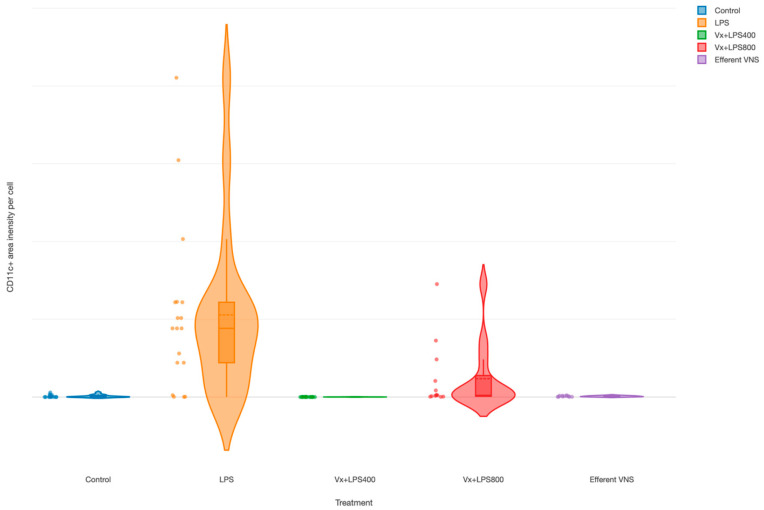
Fetal terminal ileum inflammatory response to LPS and vagus nerve manipulation: impact of vagal denervation (vagotomy) and efferent vagus nerve stimulation (VNS) on the M1 (CD11c+ cells) macrophage behavior. CD11c total area normalized by the cell count (pixels). GLM: increase; main effect—LPS and Vx + LPS800. Control (n = 5), LPS (n = 12), Vx + LPS400 (n = 6), Vx + LPS800 (n = 4), efferent VNS (n = 5).

**Figure 5 biology-13-00038-f005:**
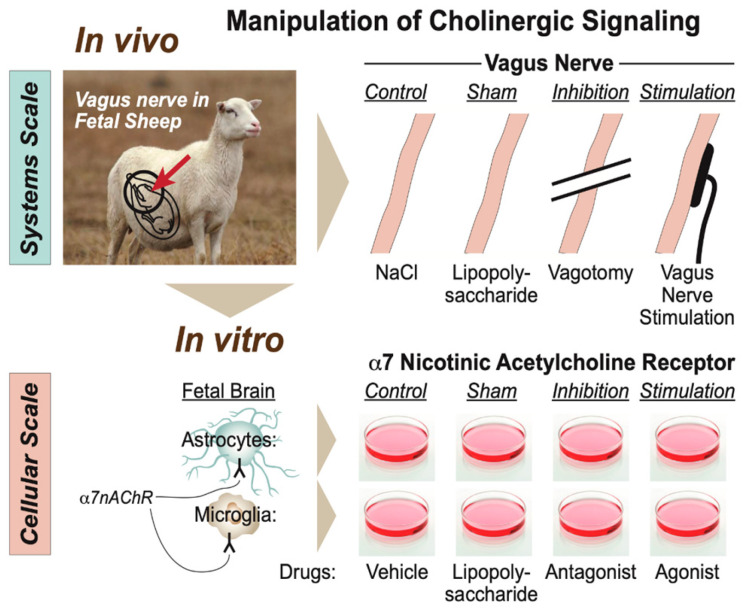
Modeling physiological scales of the organization of cholinergic signaling. (Adopted from [43] with permission).

**Table 1 biology-13-00038-t001:** Arterial blood gases and metabolites during the lipopolysaccharide (LPS)-induced fetal inflammatory response and vagus nerve manipulation (median and interquartile range, IQR).

		Experimental Group									
Parameter	Time Point	Control		LPS		Vx_LPS_400		Vx_LPS_800		Efferent VNS	
**pH**	Baseline	7.37	0.05	7.37	0.05	7.40	0.07	7.40	0.01	7.38	0.03
	1 h	7.37	0.04	7.36	0.07	7.39	0.06	7.38	0.04	7.37	0.05
	**3 h**	7.36	0.06	7.34	0.09	7.36	0.08	7.36	0.01	7.35	0.02
	**6 h**	7.37	0.04	7.35	0.08	7.35	0.07	7.35	0.03	7.36	0.07
	24 h	7.36	0.05	7.36	0.04	7.39	0.06	7.40	0.03	7.37	0.03
	27 h	7.33	0.08	7.35	0.03	7.36	0.05	7.39	0.03	7.37	0.04
	30 h	7.35	0.03	7.35	0.03	7.35	0.06	7.40	0.03	7.37	0.05
	**48 h**	7.34	0.07	7.35	0.07	7.37	0.03	7.38	0.02	7.37	0.03
	**54 h**	7.34	0.07	7.35	0.04	7.36	0.06	7.37	0.02	7.36	0.04
	Median IQR	7.36	0.05	**7.35**	**0.06**	7.37	0.06	**7.38**	**0.03**	7.36	0.04
**paO2**	Baseline	15.8	13.0	19.6	10.6	20.2	5.7	21.3	6.5	19.7	9.6
	1 h	17.2	16.2	19.0	8.5	20.6	6.0	24.0	1.4	18.6	7.8
	3 h	18.0	18.3	19.5	8.6	19.5	6.3	22.1	2.7	18.5	8.1
	6 h	18.4	19.5	17.3	8.5	17.7	9.3	19.4	4.6	16.9	8.9
	24 h	18.8	14.1	18.8	9.8	19.4	6.5	21.3	0.7	18.7	10.0
	27 h	18.2	7.0	20.4	7.2	20.9	9.8	22.9	2.4	17.9	7.0
	30 h	18.0	5.5	18.8	7.5	19.4	9.3	20.9	2.6	18.3	12.1
	48 h	17.9	11.1	18.2	9.3	20.7	8.9	18.7	11.6	19.2	8.9
	54 h	18.8	12.2	21.2	9.5	21.3	6.0	19.3	11.9	18.0	8.3
	Median IQR	17.9	13.0	**19.2**	**8.8**	20.0	7.5	21.1	4.9	18.4	9.0
**paCO2**	Baseline	51.2	6.2	51.6	6.9	48.6	8.4	49.4	3.3	49.9	2.4
	1 h	52.0	11.0	51.5	11.3	49.0	6.2	49.6	4.5	50.3	4.9
	3 h	51.3	10.2	51.6	11.3	51.0	8.9	52.0	2.3	51.4	4.6
	**6 h**	52.8	12.1	53.1	8.5	53.0	9.9	54.7	0.8	52.6	6.3
	24 h	52.9	11.6	51.9	7.2	50.4	6.5	51.0	2.3	51.6	5.4
	27 h	56.5	1.1	50.7	5.8	50.0	11.3	49.5	2.7	52.7	5.5
	30 h	56.3	1.1	51.3	5.0	53.0	7.1	49.6	6.0	52.4	4.5
	48 h	52.8	6.0	52.0	6.1	49.7	7.1	50.5	5.5	52.1	4.9
	**54 h**	54.5	11.9	50.5	10.5	52.6	6.3	51.9	3.9	52.6	5.3
	Median IQR	53.3	7.9	51.6	8.1	50.8	8.0	**50.9**	**3.5**	51.8	4.9
**O2Sat**	Baseline	44.8	32.7	52.5	29.3	50.4	21.9	60.4	10.2	54.6	35.5
	1 h	48.1	44.1	47.9	23.7	50.7	26.0	64.8	9.2	49.3	29.7
	3 h	53.9	55.5	51.1	33.0	47.4	21.1	54.9	7.2	47.5	26.9
	**6 h**	44.9	55.9	41.9	25.6	40.3	36.4	43.2	13.1	41.2	40.3
	24 h	44.4	29.1	49.1	31.8	49.9	26.2	52.3	0.8	48.9	38.0
	27 h	35.0	8.3	53.2	23.6	57.2	34.9	58.5	8.7	45.7	24.8
	30 h	37.8	8.1	47.4	29.8	51.5	31.8	52.0	10.6	46.7	44.2
	48 h	41.0	24.1	45.2	19.9	49.4	34.1	42.2	26.4	49.4	34.6
	54 h	44.0	38.0	48.7	23.4	51.2	23.6	44.4	18.3	44.9	27.4
	Median IQR	43.8	32.9	48.6	26.7	49.8	28.4	**52.5**	**11.6**	47.6	33.5
**Lactate**	Baseline	1.4	2.1	1.4	1.1	1.6	0.6	1.8	0.9	1.7	0.8
	1 h	1.4	2.3	1.6	1.8	1.6	0.8	2.0	0.6	1.7	0.7
	**3 h**	1.8	2.9	2.2	2.4	2.0	0.6	3.1	1.2	2.6	1.4
	**6 h**	2.0	3.0	2.2	2.4	2.4	0.6	3.9	1.4	3.6	2.1
	24 h	1.3	0.9	1.3	0.8	1.2	0.7	2.2	0.6	1.8	1.5
	27 h	3.0	0.8	1.5	0.6	1.1	1.5	2.3	0.6	1.7	1.8
	30 h	2.8	0.3	1.4	1.4	1.0	0.8	2.0	0.6	1.7	2.0
	48 h	1.4	1.9	1.3	0.9	1.6	1.1	2.0	0.4	1.6	1.0
	54 h	1.5	1.7	1.3	1.0	1.5	1.2	2.0	0.3	1.7	0.9
	Median IQR	1.9	1.8	1.6	1.4	1.6	0.9	2.4	0.7	2.0	1.4
**Glucose**	Baseline	14.0	7.5	17.5	9.4	20.0	14.6	19.0	10.5	17.5	6.4
	1 h	14.0	7.5	16.5	10.5	21.0	16.5	19.0	4.5	16.0	8.6
	3 h	15.5	6.8	15.5	9.0	21.0	7.5	18.0	4.5	13.5	12.4
	6 h	15.5	8.6	17.0	9.0	25.0	13.5	18.0	4.5	17.5	7.1
	24 h	15.0	6.0	17.0	7.1	17.0	10.1	24.0	4.5	18.0	17.6
	27 h	22.5	12.8	16.0	5.6	17.0	6.0	21.0	9.0	19.5	16.1
	30 h	22.5	8.3	17.0	9.8	19.0	7.5	21.5	7.9	20.0	16.1
	48 h	16.0	10.9	17.0	4.9	21.5	11.3	23.0	16.5	18.0	9.8
	54 h	20.0	21.8	17.0	8.3	21.5	6.0	23.0	12.0	18.0	8.6
	Median IQR	17.2	10.0	16.7	8.2	**20.3**	**10.3**	**20.7**	**8.2**	17.6	11.4
**BE**	Baseline	3.7	5.9	2.8	3.5	4.3	0.7	4.6	2.4	4.1	2.2
	1 h	3.2	5.0	2.8	2.9	4.0	1.7	2.9	2.7	3.3	2.1
	**3 h**	3.0	4.6	1.8	3.4	2.9	2.0	2.5	2.5	2.2	3.1
	6 h	3.2	6.9	2.9	4.0	4.0	2.7	3.3	2.6	3.9	3.0
	24 h	3.9	9.0	3.2	2.1	5.3	1.6	4.4	4.5	4.9	1.9
	27 h	3.7	6.3	2.1	1.5	3.2	3.8	3.8	3.4	4.5	3.4
	30 h	4.9	2.6	2.6	1.4	4.8	1.6	3.3	2.3	3.8	2.6
	48 h	3.2	5.2	1.7	4.4	4.5	3.9	2.9	2.4	2.9	3.1
	**54 h**	2.9	4.9	1.6	3.9	4.4	2.0	3.8	2.0	2.8	3.1
	Median IQR	3.5	5.6	**2.4**	**3.0**	4.1	2.2	3.5	2.8	3.6	2.7

Bold where GLM (vs. Control group & vs. Baseline) *p* < 0.05.

**Table 2 biology-13-00038-t002:** Cardiovascular responses during the lipopolysaccharide (LPS)-induced fetal inflammatory response and vagus nerve manipulation (median and interquartile range, IQR).

		Experimental Group									
Parameter	Time Point	Control		LPS		Vx_LPS_400		Vx_LPS_800		Efferent VNS	
**FHR**	Baseline	165	34	155	29	149	24	148	21	159	27
bpm	1 h	153	51	155	39	146	26	161	32	155	32
	3 h	150	37	156	24	158	8	156	44	152	20
	**6 h**	162	47	179	49	178	26	169	16	174	12
	24 h	158	75	158	43	139	22	147	43	172	45
	27 h	186	0	162	71	144	26	150	29	160	27
	30 h			167	26	147	29	155	40	160	35
	**48 h**	161	18	168	46	151	39	161	48	172	30
	54 h	135	53	153	45	153	27	167	37	160	33
	Median IQR	159	39	161	41	152	25	157	34	163	29
**dBP**	Baseline	37	14	39	6	37	6	32	18	40	9
mmHg	1 h	38	18	37	13	39	2	39	31	41	18
	3 h	40	15	36	8	39	6	37	1	36	13
	**6 h**	32	18	33	9	34	6	32	14	32	9
	24 h	39	12	34	11	41	8	34	6	37	12
	27 h	45	0	31	4	40	12	37	6	41	10
	**30 h**			31	8	40	5	35	18	33	9
	48 h	35	8	35	6	31	5	45	7	34	10
	54 h	36	7	34	10	36	8	49	2	36	14
	Median IQR	38	11	**34**	**8**	37	6	38	12	37	12
**mBP**	Baseline	46	14	44	5	44	4	42	16	47	11
mmHg	1 h	43	20	44	11	46	7	44	33	46	15
	3 h	45	10	43	11	47	14	42	8	45	12
	**6 h**	41	16	40	9	43	14	40	11	41	14
	24 h	46	11	41	15	46	6	42	10	44	7
	27 h	51	0	39	6	47	15	44	9	44	5
	30 h			38	10	46	12	43	16	45	8
	48 h	45	6	41	6	40	2	53	12	42	11
	54 h	43	4	39	6	44	9	59	1	42	8
	Median IQR	45	10	41	9	45	9	45	13	44	10
**sBP**	Baseline	54	15	50	9	49	17	54	12	58	16
mmHg	1 h	52	19	51	14	54	6	53	33	54	14
	3 h	48	12	51	13	55	3	50	11	51	13
	**6 h**	51	13	45	12	53	24	47	9	51	18
	24 h	50	8	49	14	51	6	52	16	51	5
	27 h	57	0	45	13	55	18	51	11	50	8
	30 h			45	13	53	20	52	13	54	12
	48 h	56	27	46	12	49	2	62	22	52	13
	54 h	48	5	45	7	53	7	70	1	50	13
	Median IQR	52	13	47	12	52	12	**54**	**14**	**52**	**12**

FHR, fetal heart rate in beats per minute (bpm); dBP, diastolic fetal arterial blood pressure (mmHg); mBP, mean fetal arterial blood pressure (mmHg); sBP, systolic fetal arterial blood pressure (mmHg).

**Table 3 biology-13-00038-t003:** Integrative overview of the study’s findings (cf. visual abstract).

System	Modality	GLM (vs. Control and Baseline Time Point)				Correlation	Comment
		LPS	Vx + LPS400	Vx + LPS800	Efferent VNS		
**General**	**Blood gas**	None					Mild sepsis without overt cardiovascular decompensation; note the habituation effect.
	**Lactate**	1.5x↑ at 3 and 6 h					
**Cardiovascular**	**FHR**	↑ at 6 h and 48 h with habituation					
	**dBP**	↓ 6 and 30 h					
	**mBP**	↓ at 6 h					
	**sBP**	↓ 6 h		↑			
**Metabolic**	**Glucose**		~1.3× ↑			Control > eff VNS > Vx	Vx triggers a chronic rise in glucose levels, starting prior to and regardless of LPS; eff. VNS restores this.
	**Insulin**		↑ at 3 h; Vx > eff VNS				Vx causes transient hyperinsulinemia at the IL-6 peak, but not at the baseline; eff. VNS restores this in part; Vx disrupts glucose–insulin homeostasis.
**Inflammation**	**IL-6** (systemic arterial)	↑ 3 and 6 h	↓ to control levels	↑ at 3 and 6 h; Vx > LPS > eff VNS		None with insulin	Vx + LPS400 is surprisingly anti-inflammatory; Vx brake on inflammatory response is LPS dose-dependent and ablates the memory (habituation to the 2nd hit); VNS reduces the inflammatory response and may boost the memory effect.
	**CD11c** (terminal ileum)	↑		↑			Similar to the systemic response, the LPS dose-dependent M1 inflammatory response in the ileum was restored with eff. VNS; we could not validate insulin’s anti-inflammatory role via direct correlations to systemic or regional inflammation.

## Data Availability

Data are contained within the article.

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
