# Peer review of "The Vagus Nerve Regulates Immunometabolic Homeostasis in the Ovine Fetus near Term: The Impact on Terminal Ileum"

_biology, 2024, doi:10.3390/biology13010038_

Round 1

Reviewer 1 Report

Comments and Suggestions for Authors

Line 103, the citation in the sentence, “In humans, Thayer and Sternberg (2006) reviewed the evidence from four studies in” shall be replaced with numerical format instead of Author: Year format.

A separate conclusion section at the end of the article highlighting the key findings of the study if included will be more useful for a clear understanding of the subject by the audience.

The acronyms like, “BE”, “LPS” etc. used in the tables, figures and text shall be either elaborated in the first mention or a separate section may be allocated to the abbreviations used in the article. This will help the readers for an instant understanding of the multiple acronyms used in the article.

Overall, the article is well written, rationale of the study is clearly highlighted in the introduction section, objectives of the study has been achieved by the authors by extensive experimental work, results are in sufficient details and the findings are clearly interpreted in the discussion section.

Author Response

Line 103, the citation in the sentence, “In humans, Thayer and Sternberg (2006) reviewed the evidence from four studies in” shall be replaced with numerical format instead of Author: Year format.

Response: Thank you. Done. 

A separate conclusion section at the end of the article highlighting the key findings of the study if included will be more useful for a clear understanding of the subject by the audience.

Response: Thank you. We have done so by providing the subsection “Significance and Perspectives” at the end of the Discussion section. Moreover, we also provide a visual abstract summarizing the findings.

The acronyms like, “BE”, “LPS” etc. used in the tables, figures and text shall be either elaborated in the first mention or a separate section may be allocated to the abbreviations used in the article. This will help the readers for an instant understanding of the multiple acronyms used in the article.

Response: Thank you. All acronyms are now spelled out upon first use as is the standard practice. Thank you for catching the missing first spellings of BE and LPS. This has been corrected. 

Overall, the article is well written, rationale of the study is clearly highlighted in the introduction section, objectives of the study has been achieved by the authors by extensive experimental work, results are in sufficient details and the findings are clearly interpreted in the discussion section.

Response: Thank you for these comments!

Reviewer 2 Report

Comments and Suggestions for Authors

Dear author, 

thanks for your manuscript submission. please find major revision:

General Comments:

1. Introduction:

    - Provide a more detailed background on the current understanding of glucosensing elements and their relation to the vagus nerve.

    - Clearly state the knowledge gaps and the significance of addressing them.

    - Consider integrating relevant citations to support the importance of investigating the physiological role of the vagus nerve in glucosensing and its potential link to fetal inflammation.

2. Hypothesis:

    - Clearly articulate the rationale behind the hypothesis. Why would vagotomy lead to increased systemic glucose levels? What is the expected relationship between vagus nerve stimulation and glucose levels during inflammation?

3. Methods:

    - Clarify the rationale behind choosing near-term fetal sheep as the experimental model.

    - Specify the criteria for selecting the sample size (n=57) and discuss any potential limitations associated with this sample size.

    - Clearly describe the surgical procedures, especially the vagotomy and VNS administration, to ensure reproducibility.

    - Explain the choice of the LPS dose and why some animals received a double dose (LPS800).

4. Results:

    - Provide more context for interpreting the blood gas and cardiovascular changes observed, particularly in the context of mild septicemia.

    - Clearly present the results in a structured manner, using tables or figures to enhance clarity.

    - Discuss the potential implications of the M1 macrophage activity observed in the LPS and Vx+LPS800 groups. What does this mean in the context of fetal inflammation?

    - Discuss any unexpected findings and potential sources of bias or confounding factors.

5. Discussion:

    - Summarize the key findings concisely and discuss their implications for the understanding of vagal innervation and its role in glucose regulation during inflammation.

    - Compare the results with existing literature and discuss any inconsistencies or novel findings.

    - Consider potential alternative explanations for the observed effects and address their feasibility.

    - Discuss the broader implications of the study and potential avenues for future research.

6. Conclusion:

    - Provide a clear and concise summary of the main findings and their significance.

Specific Comments:

    - Specify the types of glucosensing elements and briefly mention their known functions. Provide a clear transition to the role of the vagus nerve in relaying glucose information.

    - Mention specific challenges or gaps in knowledge regarding the physiological role of the vagus nerve in glucosensing.

    - Introduce key terms such as "hormetic" and provide brief explanations to ensure readers' understanding.

    - Elaborate on why vagotomy is expected to trigger a rise in systemic glucose levels and how this relates to inflammation.

    - Clarify the surgical preparation process and its relevance to the study objectives.

    - Provide more details on the selection criteria for the fetal sheep and discuss any potential biases associated with this choice.

    - Specify the criteria for choosing the LPS dose, especially the rationale for selecting 400 ng/fetus/day and the doubling of the dose in the Vx+LPS800 group.

    - Clarify the reasons for choosing a 72-hour postoperative period before starting the experiment and discuss any potential implications for the results.

    - Clearly present the blood gas and cardiovascular changes, providing context for their significance in the study.

    - Use tables or figures to present the results in a more organized and reader-friendly manner.

    - Discuss the implications of the M1 macrophage activity observed in the LPS and Vx+LPS800 groups in the context of fetal inflammation.

    - Clearly present and discuss the glucose and insulin concentrations in the Vx+LPS group, emphasizing the significance of the 1.3-fold and 2.3-fold changes.

    - Summarize the key findings and discuss their implications for the study's objectives.

    - Compare the study results with existing literature, highlighting any consistencies or discrepancies.

    - Address potential alternative explanations for the observed effects and discuss their feasibility.

    - Discuss the broader implications of the study and propose potential directions for future research.

    - Provide a clear and concise summary of the main findings and their significance.

    - Ensure consistent formatting and citation style throughout the manuscript.

    - Proofread the manuscript for grammatical errors and clarity of expression.

Note:

Remember to adjust the line numbers and content specifics based on the actual manuscript text. These comments are meant to guide you in revising and improving the clarity, structure, and presentation of your manuscript.

Comments on the Quality of English Language

Moderate editing of English language required

Author Response

General Comments:

  1. Introduction:

    - Provide a more detailed background on the current understanding of glucosensing elements and their relation to the vagus nerve.

Response: Thank you. This is done in detail on lines 10-17 and 28-57. Fifteen relevant reviews and studies are cited. A more detailed account is out of scope of this original research article. 

    - Clearly state the knowledge gaps and the significance of addressing them.

Response: Thank you. 

    - Consider integrating relevant citations to support the importance of investigating the physiological role of the vagus nerve in glucosensing and its potential link to fetal inflammation.

 Response: Thank you. Fifteen relevant reviews and studies are cited including the following ones:

  1. Kao, L.S.; Morris, B.H.; Lally, K.P.; Stewart, C.D.; Huseby, V.; Kennedy, K.A. Hyperglycemia and Morbidity and Mortality in Extremely Low Birth Weight Infants. J. Perinatol. 2006, 26, 730–736, doi:10.1038/sj.jp.7211593.
  2. van der Lugt, N.M.; Smits-Wintjens, V.E.H.J.; van Zwieten, P.H.T.; Walther, F.J. Short and Long Term Outcome of Neonatal Hyperglycemia in Very Preterm Infants: A Retrospective Follow-up Study. BMC Pediatr. 2010, 10, 52, doi:10.1186/1471-2431-10-52.
  3. Hall, N.J.; Peters, M.; Eaton, S.; Pierro, A. Hyperglycemia Is Associated with Increased Morbidity and Mortality Rates in Neonates with Necrotizing Enterocolitis. J. Pediatr. Surg. 2004, 39, 898–901; discussion 898–901.
  4. LeBlanc, M.H.; Huang, M.; Vig, V.; Patel, D.; Smith, E.E. Glucose Affects the Severity of Hypoxic-Ischemic Brain Injury in Newborn Pigs. Stroke 1993, 24, 1055–1062.
  5. Liu, H.; Zhan, P.; Meng, F.; Wang, W. Chronic Vagus Nerve Stimulation for Drug-Resistant Epilepsy May Influence Fasting Blood Glucose Concentration. Biomed. Eng. Online 2020, 19, 40, doi:10.1186/s12938-020-00784-1.

  1. Hypothesis:

    - Clearly articulate the rationale behind the hypothesis. Why would vagotomy lead to increased systemic glucose levels? What is the expected relationship between vagus nerve stimulation and glucose levels during inflammation?

 Response: Thank you. We appreciate the thoughtful questions.

  1. A) Why would Vx increase systemic glucose levels? 
  • We observe on Lines 44-47 from past studies that 

reduced vagal activity is a hallmark of diabetes.[21–24][21–24] Moreover, chronic cervical vagus nerve stimulation (VNS) for control of drug-resistant epilepsy increases their fasting glucose levels, albeit still within healthy limits.[25][25]

  1. B) What is the expected relationship between vagus nerve stimulation and glucose levels during inflammation?
  • This is a tougher question to answer fully due to the lack of clear data as we discuss. That is what makes it an interesting research question to study. We observe on Lines 28-32 that

The role of glycemic control in the regulation of the inflammatory response is not well understood with evidence accruing that hyperglycemia exacerbates the inflammation induced by endotoxin.[16][16] Similar assumptions are made for the etiology of NEC and other major neonatal causes of morbidity and mortality, especially in premature neonates.[17,18][17,18],[19,20]

  • Summarizing the evidence, we state on lines 58-71 that [key points bolded for easy review here]

Taken together, it is evident that our knowledge about vagal glycemic control across different species and the developmental stages of this control is still very limited. However, it is clear that the vagus nerve exerts - selectively via its efferent and afferent pathways - a powerful modulatory influence on glucose and inflammatory homeostatic control systems in health and disease. Evidence is also compelling that the vagus nerve’s regulatory role is present already during fetal development, a period when adverse exposures are known to have powerful long-lasting reprogramming effects on postnatal health and predisposition to disease in later life.[8,29,30][8,29,30]

Consequently, we hypothesized that 1) bilateral cervical vagotomy in a mature, near-term, fetal sheep will cause an increase in systemic glucose levels; 2) this increase will be correlated to a higher degree of systemic inflammation and the inflammation in the terminal ileum, NEC’s locus minoris resistentiae; 3) selective efferent VNS will reverse these patterns (glycemic control hypothesis).

  1. Methods:

    - Clarify the rationale behind choosing near-term fetal sheep as the experimental model.

 Response: Thank you. Pregnant sheep is the best animal model of fetal physiology for studying complex systems on multiple levels, from cell to integrative physiology. It has served the scientific community since early 1960ies and informed much of the present practical obstetrical knowledge about fetal development, especially cardiovascular and brain physiologies.[8] We state that on Lines 5-9 in the Introduction.

We reported the detailed approach including Vx and VNS elsewhere.[7,31] These publications are open-source and detail the rationale.

  1. Castel, A.; Burns, P.M.; Benito, J.; Liu, H.L.; Kuthiala, S.; Durosier, L.D.; Frank, Y.S.; Cao, M.; Paquet, M.; Fecteau, G.; et al. Recording and Manipulation of Vagus Nerve Electrical Activity in Chronically Instrumented Unanesthetized near Term Fetal Sheep. J. Neurosci. Methods 2021, 109257, doi:10.1016/j.jneumeth.2021.109257.
  2. Morrison, J.L.; Berry, M.J.; Botting, K.J.; Darby, J.R.T.; Frasch, M.G.; Gatford, K.L.; Giussani, D.A.; Gray, C.L.; Harding, R.; Herrera, E.A.; et al. Improving Pregnancy Outcomes in Humans through Studies in Sheep. Am. J. Physiol. Regul. Integr. Comp. Physiol. 2018, doi:10.1152/ajpregu.00391.2017.
  3. Burns, P.; Liu, H.L.; Kuthiala, S.; Fecteau, G.; Desrochers, A.; Durosier, L.D.; Cao, M.; Frasch, M.G. Instrumentation of Near-Term Fetal Sheep for Multivariate Chronic Non-Anesthetized Recordings. J. Vis. Exp. 2015, e52581, doi:10.3791/52581.

    - Specify the criteria for selecting the sample size (n=57) and discuss any potential limitations associated with this sample size.

Response: Thank you. We described the composition of each experimental group on Lines 127-140. We strived to have at least 5-7 animals in each group, as is custom in such studies. The LPS group achieved higher numbers due to titrating the LPS dose in the first three animals as stated (Line 128). Additional five animals received the double LPS dose of 800 ng/fetus/day as stated (Line 129) to match the Vx+LPS800 cohort (Line 136-137). That was because we discovered the hormetic effect in Vx+LPS400 versus Vx+LPS800 groups as we report in Results (Lines 333-336). We report raw data in all figures (2-4) along with statistical findings which we suggest speak for themselves. 

    - Clearly describe the surgical procedures, especially the vagotomy and VNS administration, to ensure reproducibility.

Response: Thank you. We reported the detailed approach including Vx and VNS elsewhere.[7,31] These publications are open-source, include video instructions and detail the exact step-wise approach. We do not repeat this text here to avoid self-plagiarism.

    - Explain the choice of the LPS dose and why some animals received a double dose (LPS800).

 Response: Thank you. The LPS group achieved higher numbers due to titrating the LPS dose in the first three animals as stated (Line 128). Additional five animals received the double LPS dose of 800 ng/fetus/day as stated (Line 129) to match the Vx+LPS800 cohort (Line 136-137). That was because we discovered the hormetic effect in Vx+LPS400 versus Vx+LPS800 groups as we report in Results (Lines 333-336). We report this on Lines 127-140 and Lines 333-336.

  1. Results:

    - Provide more context for interpreting the blood gas and cardiovascular changes observed, particularly in the context of mild septicemia.

Response: Thank you. Mild septicemia changes with a 40% elevation in lactate are compatible with the basic understanding of the pathophysiology of sepsis as found in text books, so we felt it not necessary to elaborate on this. A simple Google Scholar search can suffice for an interested reader to dive deeper.

The same is true for the reported cardiovascular changes with tachycardia and hypotension. If this reviewer wishes to elaborate on any additional context that we should provide, we will be happy to follow up.

    - Clearly present the results in a structured manner, using tables or figures to enhance clarity.

Response: Thank you. We have done so. The Results section is subdivided into subsections and figures and tables report the respective data.

    - Discuss the potential implications of the M1 macrophage activity observed in the LPS and Vx+LPS800 groups. What does this mean in the context of fetal inflammation?

Response: Thank you. As we state on Lines 388-389, the findings in for M1 macrophages in the LPs group were as we had reported elsewhere (Ref. 35). The new findings on the effects of Vx+LPS on M1 macrophages for LPS800 group are compatible and consistent with the pro-inflammatory effect of this treatment observed with other biomarkers as well. It seems that this exposure promotes the inflammatory response. We discuss this on Lines 414-419 and 521-527.

    - Discuss any unexpected findings and potential sources of bias or confounding factors.

 Response: Thank you. We do indeed focus in this manuscript on the unexpected finding of hormesis dedicating to this the two Subsection on Lines 406-440 and Lines 520-549.

The findings are summarized in Table 3 to facilitate the systematic interpretation.

We discuss limitations on Lines 507-513 as well as 545-549.

  1. Discussion:

    - Summarize the key findings concisely and discuss their implications for the understanding of vagal innervation and its role in glucose regulation during inflammation.

Response: Thank you. We address this requirement in four complementary ways.

First, we summarize briefly the key findings on Lines 400-405. 

Second, we provide these findings in Table 3 to facilitate the systematic interpretation.

Third, we synthesize the insights in the subsection “Significance and perspectives” (Lines 571-594).

Fourth, we provide a professionally designed and created visual abstract summarizing the findings in a graphics format.

We discuss the implications  for the understanding of vagal innervation and its role in glucose regulation during inflammation in the dedicated Subsestions “Immunometabolic effects of vagus nerve manipulation”, “Hormesis” and “Implications for NEC etiology and avenues of treatment”.

    - Compare the results with existing literature and discuss any inconsistencies or novel findings.

Response: Thank you. This is done throughout the manuscript. Of note, the surprising finding of hormesis is discussed in detail on Lines 520-549. 32 manuscripts are cited in this work representing the work of other authors/teams.

    - Consider potential alternative explanations for the observed effects and address their feasibility.

Response: Thank you. We have done so, e.g., Lines 456-458, Lines 545-549, Lines 565-570.

    - Discuss the broader implications of the study and potential avenues for future research.

Response: Thank you. We do that in the subsections on Lines 550-594.

  1. Conclusion:

    - Provide a clear and concise summary of the main findings and their significance.

Response: Thank you. We do that in the subsection “Significance and perspectives” (Lines 571-594).

Specific Comments:

    - Specify the types of glucosensing elements and briefly mention their known functions. Provide a clear transition to the role of the vagus nerve in relaying glucose information.

    - Mention specific challenges or gaps in knowledge regarding the physiological role of the vagus nerve in glucosensing.

    - Introduce key terms such as "hormetic" and provide brief explanations to ensure readers' understanding.

    - Elaborate on why vagotomy is expected to trigger a rise in systemic glucose levels and how this relates to inflammation.

    - Clarify the surgical preparation process and its relevance to the study objectives.

    - Provide more details on the selection criteria for the fetal sheep and discuss any potential biases associated with this choice.

    - Specify the criteria for choosing the LPS dose, especially the rationale for selecting 400 ng/fetus/day and the doubling of the dose in the Vx+LPS800 group.

    - Clarify the reasons for choosing a 72-hour postoperative period before starting the experiment and discuss any potential implications for the results.

    - Clearly present the blood gas and cardiovascular changes, providing context for their significance in the study.

    - Use tables or figures to present the results in a more organized and reader-friendly manner.

    - Discuss the implications of the M1 macrophage activity observed in the LPS and Vx+LPS800 groups in the context of fetal inflammation.

    - Clearly present and discuss the glucose and insulin concentrations in the Vx+LPS group, emphasizing the significance of the 1.3-fold and 2.3-fold changes.

    - Summarize the key findings and discuss their implications for the study's objectives.

    - Compare the study results with existing literature, highlighting any consistencies or discrepancies.

    - Address potential alternative explanations for the observed effects and discuss their feasibility.

    - Discuss the broader implications of the study and propose potential directions for future research.

    - Provide a clear and concise summary of the main findings and their significance.

    - Ensure consistent formatting and citation style throughout the manuscript.

    - Proofread the manuscript for grammatical errors and clarity of expression.

Response: Thank you very much for these thoughtful comments. We hope to have addressed them in the above responses.

Round 2

Reviewer 2 Report

Comments and Suggestions for Authors

Dear author,

After careful revision, manuscript revised successfully, and can be proceed further for publication.

Comments on the Quality of English Language

Minor editing of English language required

Author Response

Dear Reviewer,

thank you very much for your review, once again, and for the comments to perform minor editing of English. We have done that now and resubmit the revised manuscript herewith in the hope it will be found satisfactory for publication.

With best regards,

Dr. Martin Frasch